# Adaptive Neural Compilation

**Rudy Bunel**\*
University of Oxford
rudy@robots.ox.ac.uk

**Alban Desmaison**\*
University of Oxford
alban@robots.ox.ac.uk

**Pushmeet Kohli**
Microsoft Research
pkohli@microsoft.com

**Philip H.S. Torr**
University of Oxford
philip.torr@eng.ox.ac.uk

**M. Pawan Kumar**
University of Oxford
pawan@robots.ox.ac.uk

## Abstract

This paper proposes an adaptive neural-compilation framework to address the problem of learning efficient programs. Traditional code optimisation strategies used in compilers are based on applying pre-specified set of transformations that make the code faster to execute without changing its semantics. In contrast, our work involves adapting programs to make them more efficient while considering correctness only on a target input distribution. Our approach is inspired by the recent works on differentiable representations of programs. We show that it is possible to compile programs written in a low-level language to a differentiable representation. We also show how programs in this representation can be optimised to make them efficient on a target input distribution. Experimental results demonstrate that our approach enables learning *specifically-tuned* algorithms for given data distributions with a high success rate.

## 1   Introduction

Algorithm design often requires making simplifying assumptions about the input data. Consider, for instance, the computational problem of accessing an element in a linked list. Without the knowledge of the input data distribution, one can only specify an algorithm that runs in a time linear in the number of elements of the list. However, suppose all the linked lists that we encountered in practice were ordered in memory. Then it would be advantageous to design an algorithm specifically for this task as it can lead to a constant running time. Unfortunately, the input data distribution of a real world problem cannot be easily specified as in the above simple example. The best that one can hope for is to obtain samples drawn from the distribution. A natural question that arises from these observations: "How can we adapt a generic algorithm for a computational task using samples from an unknown input data distribution?"

The process of finding the most efficient implementation of an algorithm has received considerable attention in the theoretical computer science and code optimisation community. Recently, Conditionally Correct Superoptimization [14] was proposed as a method for leveraging samples of the input data distribution to go beyond semantically equivalent optimisation and towards data-specific performance improvements. The underlying procedure is based on a stochastic search over the space of all possible programs. Additionally, they restrict their applications to reasonably small, loop-free programs, thereby limiting their impact in practice.

In this work, we take inspiration from the recent wave of machine-learning frameworks for estimating programs. Using recurrent models, Graves et al. [2] introduced a fully differentiable representation of a program, enabling the use of gradient-based methods to learn a program from examples. Many other models that have been published recently [3, 5, 6, 8] build and improve on the early work by Graves

et al. [2]. Unfortunately, these models are usually complex to train and need to rely on methods such as curriculum learning or gradient noise to reach good solutions as shown by Neelakantan et al. [10]. Moreover, their interpretability is limited. The learnt model is too complex for the underlying algorithm to be recovered and transformed into a regular computer program.

The main focus of the machine-learning community has thus far been on learning programs from scratch, with little emphasis on running time. However, for nearly all computational problems, it is feasible to design generic algorithms for the worst-case. We argue that a more pragmatic goal for the machine learning community is to design methods for adapting existing programs for specific input data distributions. To this end, we propose the Adaptive Neural Compiler (ANC). We design a compiler capable of mechanically converting algorithms to a differentiable representation, thereby providing adequate initialisation to the difficult problem of optimal program learning. We then present a method to improve this compiled program using data-driven optimisation, alleviating the need to perform a wide search over the set of all possible programs. We show experimentally that this framework is capable of adapting simple generic algorithms to perform better on given datasets.

## 2 Related Works

The idea of compiling programs to neural networks has previously been explored in the literature. Siegelmann [15] described how to build a Neural Network that would perform the same operations as a given program. A compiler has been designed by Gruau et al. [4] targeting an extended version of Pascal. A complete implementation was achieved when Neto et al. [11] wrote a compiler for NETDEF, a language based on the Occam programming language. While these methods allow us to obtain an exact representation of a program as a neural network, they do not lend themselves to optimisation to improve the original program. Indeed, in their formulation, each elementary step of a program is expressed as a group of neurons with a precise topology, set of weights and biases, thereby rendering learning via gradient descent infeasible. Performing gradient descent in this parameter space would result in invalid operations and thus is unlikely to lead to any improvement. The recent work by Reed and de Freitas [12] on Neural Programmer-Interpreters (NPI) can also be seen as a way to compile any program into a neural network. It does so by learning a model that mimics the program. While more flexible than previous approaches, the NPI only learns to reproduce an existing program. Therefore it cannot be used to find a new and possibly better program.

Another approach to this learning problem is the one taken by the code optimisation community. By exploring the space of all possible programs, either exhaustively [9] or in a stochastic manner [13], they search for programs having the same results but being more efficient. The work of Sharma et al. [14] broadens the space of acceptable improvements to data-specific optimisations as opposed to the provably equivalent transformations that were previously the only ones considered. However, this method is still reliant on non-gradient-based methods for efficient exploration of the space. By representing everything in a differentiable manner, we aim to obtain gradients to guide the exploration.

Recently, Graves et al. [2] introduced a learnable representation of programs, called the Neural Turing Machine (NTM). The NTM uses an LSTM as a Controller, which outputs commands to be executed by a deterministic differentiable Machine. From examples of input/output sequences, they manage to learn a Controller such that the model becomes capable of performing simple algorithmic tasks. Extensions of this model have been proposed in [3, 5] where the memory tape was replaced by differentiable versions of stacks or lists. Kurach et al. [8] modified the NTM to introduce a notion of pointers making it more amenable to represent traditional programs. Parallel works have been using Reinforcement Learning techniques such as the REINFORCE algorithm [1, 16, 17] or Q-learning [18] to be able to work with non differentiable versions of the above mentioned models. All these models are trained only with a loss based on the difference between the output of the model and the expected output. This weak supervision results in learning becoming more difficult. For instance the Neural RAM [8] requires a high number of random restarts before converging to a correct solution [10], even when using the best hyperparameters obtained through a large grid search.

In our work, we will first show that we can design a new neural compiler whose target will be a Controller-Machine model. This makes the compiled model amenable to learning from examples. Moreover, we can use it as initialisation for the learning procedure, allowing us to aim for the more complex task of finding an efficient algorithm.

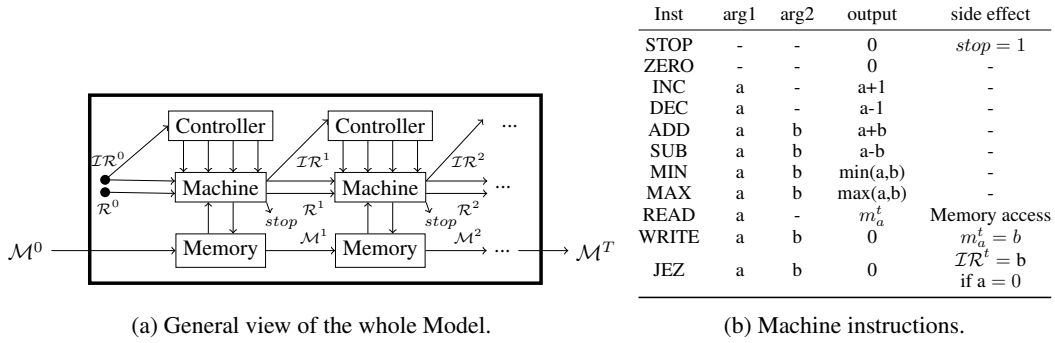

| Inst | arg1 | arg2 | output | side effect |
|---|---|---|---|---|
| STOP | - | - | 0 | $stop = 1$ |
| ZERO | - | - | 0 | - |
| INC | a | - | a+1 | - |
| DEC | a | - | a-1 | - |
| ADD | a | b | a+b | - |
| SUB | a | b | a-b | - |
| MIN | a | b | min(a,b) | - |
| MAX | a | b | max(a,b) | - |
| READ | a | - | $m_a^t$ | Memory access |
| WRITE | a | b | 0 | $m_a^t = b$ |
| JEZ | a | b | 0 | $\mathcal{IR}^t = b$ if a = 0 |

(a) General view of the whole Model.      (b) Machine instructions.

Figure 1: Model components.

## 3 Model

Our model is composed of two parts: *(i)* a Controller, in charge of specifying what should be executed; and *(ii)* a Machine, following the commands of the Controller. We start by describing the global architecture of the model. For the sake of simplicity, the general description will present a non-differentiable version of the model. Section 3.2 will then explain the modifications required to make this model completely differentiable. A more detailed description of the model is provided in the supplementary material.

### 3.1 General Model

We first define for each timestep $t$ the memory tape that contains $M$ integer values $\mathcal{M}^t = \{m_1^t, m_2^t, \ldots, m_M^t\}$, registers that contain $R$ values $\mathcal{R}^t = \{r_1^t, r_2^t, \ldots, r_R^t\}$ and the instruction register that contains a single value $\mathcal{IR}^t$. We also define a set of instructions that can be executed, whose main role is to perform computations using the registers. For example, add the values contained in two registers. We also define as a side effect any action that involves elements other than the input and output values of the instruction. Interaction with the memory is an example of such side effect. All the instructions, their computations and side effects are detailed in Figure 1b.

As can be seen in Figure 1a the execution model takes as input an initial memory tape $\mathcal{M}^0$ and outputs a final memory tape $\mathcal{M}^T$ after $T$ steps. At each step $t$, the Controller uses the instruction register $\mathcal{IR}^t$ to compute the command for the Machine. The command is a 4-tuple $e, a, b, o$. The first element $e$ is the instruction that should be executed by the Machine, enumerated as an integer. The elements $a$ and $b$ specify which registers should be used as arguments for the given instruction. The last element $o$ specifies in which register the output of the instruction should be written. For example, the command $\{ADD, 2, 3, 1\}$ means that only the value of the first register should change, following $r_1^{t+1} = ADD(r_2^t, r_3^t)$. Then the Machine will execute this command, updating the values of the memory, the registers and the instruction register. The Machine always performs two other operations apart from the required instruction. It outputs a *stop* flag that allows the model to decide when to stop the execution. It also increments the instruction register $\mathcal{IR}^t$ by one at each iteration.

### 3.2 Differentiability

The model presented above is a simple execution machine but it is not differentiable. In order to be able to train this model end-to-end from a loss defined over the final memory tape, we need to make every intermediate operation differentiable.

To achieve this, we replace every discrete value in our model by a multinomial distribution over all the possible values that could have been taken. Moreover, each hard choice that would have been non-differentiable is replaced by a continuous soft choice. We will henceforth use bold letters to indicate the probabilistic version of a value.

First, the memory tape $\mathcal{M}^t$ is replaced by an $M \times M$ matrix $\mathbf{M}^t$, where $\mathbf{M}_{i,j}^t$ corresponds to the probability of $m_i^t$ taking the value $j$. The same change is applied to the registers $\mathcal{R}^t$, replacing them with an $R \times M$ matrix $\mathbf{R}^t$, where $\mathbf{R}_{i,j}^t$ represents the probability of $r_i^t$ taking the value $j$. Finally, the instruction register is also transformed, the same way as the other registers, from a single value $\mathcal{IR}^t$ to a vector of size $M$ noted $\boldsymbol{\mathcal{IR}}^t$, where the $i$-th element represents its probability to take the value $i$.

The Machine does not contain any learnable parameter and will just execute a given command. To make it differentiable, the Machine now takes as input four probability distributions $\mathbf{e}^t$, $\mathbf{a}^t$, $\mathbf{b}^t$ and $\mathbf{o}^t$, where $\mathbf{e}^t$ is a distribution over instructions, and $\mathbf{a}^t$, $\mathbf{b}^t$ and $\mathbf{o}^t$ are distributions over registers. We compute the argument values $\mathbf{arg_1}^t$ and $\mathbf{arg_2}^t$ as convex combinations of delta-function probability distributions of the different registers values:

$$\mathbf{arg_1}^t = \sum_{i=1}^{R} \mathbf{a}_i^t \mathbf{r}_i^t \qquad \mathbf{arg_2}^t = \sum_{i=1}^{R} \mathbf{b}_i^t \mathbf{r}_i^t, \tag{1}$$

where $\mathbf{a}_i^t$ and $\mathbf{b}_i^t$ are the $i$-th values of the vectors $\mathbf{a}^t$ and $\mathbf{b}^t$. Using these values, we can compute the output value of each instruction $k$ using the following formula:

$$\forall 0 \le c \le M \quad \mathbf{out}_{k,c}^t = \sum_{0 \le i,j \le M} \mathbf{arg_1}_i^t \cdot \mathbf{arg_2}_j^t \cdot \mathbb{1}[g_k(i,j) = c \mod M], \tag{2}$$

where $g_k$ is the function associated to the $k$-th instruction as presented in Table 1b, $\mathbf{out}_{k,c}^t$ is the probability for an instruction $k$ to output the value $c$ at the time-step $t$ and $\mathbf{arg_1}_i^t$ is the probability of the argument 1 having the value $i$ at the time-step $t$. Since the executed instruction is controlled by the probability $\mathbf{e}$, the output for all instructions will also be a convex combination: $\mathbf{out}^t = \sum_{k=1}^{N} \mathbf{e}_k^t \mathbf{out}_k^t$, where $N$ is the number of instructions. This value is then stored into the registers by performing a soft-write parametrised by $\mathbf{o}^t$: the value stored in the $i$-th register at time $t+1$ is $\mathbf{r}_i^{t+1} = \mathbf{r}_i^t(1 - \mathbf{o}_i^t) + \mathbf{out}^t \mathbf{o}_i^t$, allowing the choice of the output register to be differentiable.

A special case is associated with the *stop* signal. When executing the model, we keep track of the probability that the program should have terminated before this iteration based on the probability associated at each iteration with the specific instruction that controls this flag. Once this probability goes over a threshold $\eta_{\text{stop}} \in (0, 1]$, the execution is halted. We applied the same techniques to make the side-effects differentiable, this is presented in the supplementary materials.

The Controller is the only learnable part of our model. The first learnable part is the initial values for the registers $\mathbf{R}^0$ and for the instruction register $\mathcal{IR}^0$. The second learnable part is the parameters of the Controller which computes the required distributions using:

$$\mathbf{e}^t = \mathbf{W}_e * \mathcal{IR}^t, \qquad \mathbf{a}^t = \mathbf{W}_a * \mathcal{IR}^t, \qquad \mathbf{b}^t = \mathbf{W}_b * \mathcal{IR}^t, \qquad \mathbf{o}^t = \mathbf{W}_o * \mathcal{IR}^t \tag{3}$$

where $\mathbf{W}_e$ is an $N \times M$ matrix and $\mathbf{W}_a$, $\mathbf{W}_b$ and $\mathbf{W}_o$ are $R \times M$ matrices. A representation of these matrices can be found in Figure 2c. The Controller as defined above is composed of four independent, fully-connected layers. In Section 4.3 we will see that this complexity is sufficient for our model to be able to represent *any* program.

Henceforth, we will denote by $\boldsymbol{\theta} = \{\mathbf{R}^0, \mathcal{IR}^0, \mathbf{W}_e, \mathbf{W}_a, \mathbf{W}_b, \mathbf{W}_o\}$ the set of learnable parameters.

# 4 Adaptative Neural Compiler

We will now present the Adaptive Neural Compiler. Its goal is to find the best set of weights $\boldsymbol{\theta}^*$ for a given dataset such that our model will perform the correct input/output mapping as efficiently as it can. We begin by describing our learning objective in details. The two subsequent sections will focus on making the optimisation of our learning objective computationally feasible.

## 4.1 Objective function

Our goal is to solve a given algorithmic problem efficiently. The algorithmic problem is defined as a set of input/output pairs. We also have access to a generic program that is able to perform the required mapping. In our example of accessing elements in a linked list, the transformation would consist in writing down the desired value at the specified position in the tape. The program given to us would iteratively go through the elements of the linked list, find the desired value and write it down at the desired position. If there exists some bias that would allow this traversal to be faster, we expect the program to exploit it.

Our approach to this problem is to construct a differentiable objective function that maps controller parameters to a loss. We define this loss based on the states of the memory tape and outputs of the Controller at each step of the execution. The precise mathematical formulation for each term of the loss is given in the supplementary materials. Here we present the motivation behind each of them.

**Correctness**  We first want the final memory tape to match the expected output for a given input.

**Halting**  To prevent programs from taking an infinite amount of time without stopping, we define a maximum number of iterations $T_{max}$ after which the execution is halted. Moreover, we add a penalty if the Controller didn't halt before this limit.

**Efficiency**  We penalise each iteration taken by the program where it does not stop.

**Confidence**  We finally make sure that if the Controller wants to stop, the current output is correct.

If only the correctness term was considered, nothing would encourage the learnt algorithm to halt as soon as it finished. If only correctness and halting were considered, then the program may not halt as early as possible. Confidence enables the algorithm to better evaluate when to stop.

The loss is a weighted sum of the four above-mentioned terms. We denote the loss of the $i$-th training sample, given parameters $\boldsymbol{\theta}$, as $L_i(\boldsymbol{\theta})$. Our learning objective is then specified as:

$$\min_{\boldsymbol{\theta}} \quad \sum_i L_i(\boldsymbol{\theta}) \quad \text{s.t. } \boldsymbol{\theta} \in \boldsymbol{\Theta}, \tag{4}$$

where $\Theta$ is a set over the parameters such that the outputs of the Controller, the initial values of each register and of the instruction register are all probability distributions.

The above optimisation is a highly non-convex problem. In the rest of this section, we will first present a small modification to the model that will remove the constraints to be able to use standard gradient descent-based methods. Moreover, a good initialisation is helpful to solve these non-convex problems. To alleviate this deficiency, we now introduce our Neural Compiler that will provide a good initialisation.

## 4.2  Reformulation

In order to use gradient descent methods without having to project the parameters on $\Theta$, we alter the formulation of the Controller. We use softmax layers to be able to learn learn unormalized scores that are then mapped to probability distributions. We add one after each linear layer of the Controller and for the initial values of the registers. This way, we transform the constrained-optimisation problem into an unconstrained one, allowing us to use standard gradient descent methods. As discussed in other works [10], this kind of model is hard to train and requires a high number of random restarts before converging to a good solution. We will now present a Neural Compiler that will provide good initialisations to help with this problem.

## 4.3  Neural Compiler

The goal for the Neural Compiler is to convert an algorithm, written as an unambiguous program, to a set of parameters. These parameters, when put into the controller, will reproduce the exact steps of the algorithm. This is very similar to the problem framed by Reed and de Freitas [12], but we show here a way to accomplish it *without any learning*.

The different steps of the compilation are illustrated in Figure 2. The first step is to go from the written version of the program to the equivalent list of low level instruction. This step can be seen as going from Figure 2a to Figure 2b. The illustrative example uses a fairly low-level language but traditional features of programming languages such as `loops` or `if`-statements can be supported using the JEZ instruction. The use of constants as arguments or as values is handled by introducing new registers that hold these values. The value required to be passed as target position to the JEZ instruction can be resolved at compile time.

Having obtained this intermediate representation, generating the parameters is straightforward. As can be seen in Figure 2b, each line contains one instruction, the two input registers and the output register, and corresponds to a command that the Controller will have to output. If we ensure that $\mathcal{IR}$ is a Dirac-delta distribution on a given value, then the matrix-vector product is equivalent to selecting a row of the weight matrix. As $\mathcal{IR}$ is incremented at each iteration, the Controller outputs the rows of the matrix in order. We thus have a one-to-one mapping between the lines of the intermediate representation and the rows of the weight matrix. An example of these matrices can be found in Figure 2c. The weight matrix has 10 rows, corresponding to the number of lines of code of our intermediate representation. For example, on the first line of the matrix corresponding to the output (2civ), we see that the fifth element has value 1. This is linked to the first line of code where the output of the READ operation is stored into the fifth register. With this representation, we can

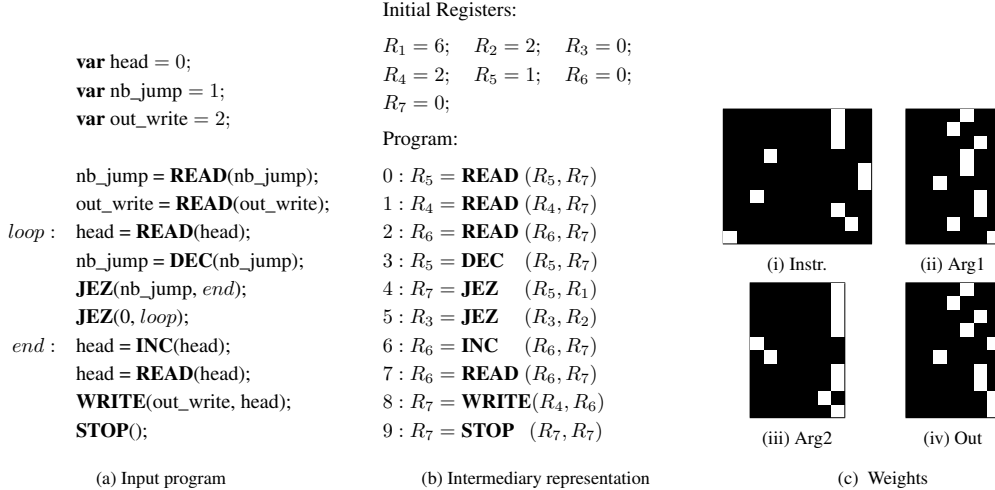

```
                        Initial Registers:

var head = 0;           R₁ = 6;   R₂ = 2;   R₃ = 0;
var nb_jump = 1;        R₄ = 2;   R₅ = 1;   R₆ = 0;
var out_write = 2;      R₇ = 0;

                        Program:
nb_jump = READ(nb_jump);    0 : R₅ = READ (R₅, R₇)
out_write = READ(out_write); 1 : R₄ = READ (R₄, R₇)
loop :  head = READ(head);   2 : R₆ = READ (R₆, R₇)
        nb_jump = DEC(nb_jump); 3 : R₅ = DEC  (R₅, R₇)
        JEZ(nb_jump, end);   4 : R₇ = JEZ  (R₅, R₁)
        JEZ(0, loop);        5 : R₃ = JEZ  (R₃, R₂)
end :   head = INC(head);    6 : R₆ = INC  (R₆, R₇)
        head = READ(head);   7 : R₆ = READ (R₆, R₇)
        WRITE(out_write, head); 8 : R₇ = WRITE(R₄, R₆)
        STOP();              9 : R₇ = STOP (R₇, R₇)

   (a) Input program     (b) Intermediary representation      (c) Weights
```

$R_1 = 6;\quad R_2 = 2;\quad R_3 = 0;$
$R_4 = 2;\quad R_5 = 1;\quad R_6 = 0;$
$R_7 = 0;$

(i) Instr.   (ii) Arg1   (iii) Arg2   (iv) Out

Figure 2: Example of the compilation process. (2a) Program written to perform the **ListK** task. Given a pointer to the head of a linked list, an integer $k$, a target cell and a linked list, write in the target cell the $k$-th element of the list. (2b) Intermediary representation of the program. This corresponds to the instruction that a Random Access Machine would need to perform to execute the program. (2c) Representation of the weights that encodes the intermediary representation. Each row of the matrix correspond to one state/line. Initial value of the registers are also parameters of the model, omitted here.

note that the number of parameters is linear in the number of lines of code in the original program and that the largest representable number in our Machine needs to be greater than the number of lines in our program.

Moreover, any program written in a regular assembly language can be rewritten to use only our restricted set of instructions. This can be done firstly because all the conditionals of the assembly language can be expressed as a combination of arithmetic and JEZ instructions. Secondly because all the arithmetic operations can be represented as a combination of our simple arithmetic operations, loops and ifs statements. This means that *any* program that can run on a regular computer, can be first rewritten to use our restricted set of instructions and then compiled down to a set of weights for our model. Even though other models use LSTM as controller, we showed here that a Controller composed of simple linear functions is expressive enough. The advantage of this simpler model is that we can now easily interpret the weights of our model in a way that is not possible if we use a recurrent network as a controller.

The most straightforward way to leverage the results of the compilation is to initialise the Controller with the weights obtained through compilation of the generic algorithm. To account for the extra softmax layer, we need to multiply the weights produced by the compiler by a large constant to output Dirac-delta distributions. Some results associated with this technique can be found in Section 5.1. However, if we initialise with exactly this sharp set of parameters, the training procedure is not able to move away from the initialisation as the gradients associated with the softmax in this region are very small. Instead, we initialise the controller with a non-ideal version of the generic algorithm. This means that the choice with the highest probability in the output of the Controller is correct, but the probability of other choices is not zero. As can be seen in Section 5.2, this allows the Controller to learn by gradient descent a new algorithm, different from the original one, that has a lower loss than the ideal version of the compiled program.

## 5   Experiments

We performed two sets of experiments. The first shows the capability of the Neural Compiler to perfectly reproduce any given program. The second shows that our Neural Compiler can adapt and improve the performance of programs. We present results of data-specific optimisation being carried out and show decreases in runtime for all the algorithms and additionally, for some algorithms, show that the runtime is a different computational-complexity class altogether. All the code required to reproduce these experiments is available online [1].

## 5.1 Compilation

The compiler described in section 4.3 allows us to go from a program written using our instruction set to a set of weights $\Theta$ for our Controller.

To illustrate this point, we implemented simple programs that can solve the tasks introduced by Kurach et al. [8] and a shortest path problem. One of these implementations can be found in Figure 2a, while the others are available in the supplementary materials. These programs are written in a specific language, and are transformed by the Neural Compiler into parameters for the model. As expected, the resulting models solve the original tasks exactly and can generalise to any input sequence.

## 5.2 ANC experiments

In addition to being able to reproduce any given program as was done by Reed and de Freitas [12], we have the possibility of optimising the resulting program further. We exhibit this by compiling program down to our model and optimising their performance. The efficiency gain for these tasks come either from finding simpler, equivalent algorithms or by exploiting some bias in the data to either remove instructions or change the underlying algorithm.

We identify three different levels of interpretability for our model: the first type corresponds to weights containing only Dirac-delta distributions, there is an exact one-to-one mapping between lines in the weight matrices and lines of assembly code. In the second type where all probabilities are Dirac-delta except the ones associated with the execution of the JEZ instruction, we can recover an exact algorithm that will use `if` statements to enumerate the different cases arising from this conditional jump. In the third type where any operation other than JEZ is executed in a soft way or use a soft argument, it is not possible to recover a program that will be as efficient as the learned one.

We present here briefly the considered tasks and biases, and report the reader to the supplementary materials for a detailed encoding of the input/output tape.

1. **Access**: Given a value $k$ and an array $A$, return $A[k]$. In the biased version, the value of $k$ is always the same, so the address of the required element can be stored in a constant. This is similar to the optimisation known as constant folding.

2. **Swap**: Given an array $A$ and two pointers $p$ and $q$, swap the elements $A[p]$ and $A[q]$. In the biased version, $p$ and $q$ are always the same so reading them can be avoided.

3. **Increment**: Given an array, increment all its element by 1. In the biased version, the array is of fixed size and the elements of the array have the same value so you do not need to read all of them when going through the array.

4. **Listk**: Given a pointer to the head of a linked list, a number $k$ and a linked list, find the value of the $k$-th element. In the biased version, the linked list is organised in order in memory, as would be an array, so the address of the $k$-th value can be computed in constant time. This is the example developed in Figure 2.

5. **Addition**: Two values are written on the tape and should be summed. No data bias is introduced but the starting algorithm is non-efficient: it performs the addition as a series of increment operation. The more efficient operation would be to add the two numbers.

6. **Sort**: Given an array $A$, sort it. In the biased version, only the start of the array might be unsorted. Once the start has been arranged, the end of the array can be safely ignored.

For each of these tasks, we perform a grid search on the loss parameters and on our hyper-parameters. Training is performed using Adam [7] and success rates are obtained by running the optimisation with 100 different random seeds. We consider that a program has been successfully optimised when two conditions are fulfilled. First, it needs to output the correct solution for all test cases presenting the same bias. Second, the average number of iterations taken to solve a problem must have decreased. Note that if we cared only about the first criterion, the methods presented in Section 5.1 would already provide a success rate of 100%, without requiring any training.

The results are presented in Table 1. For each of these tasks, we manage to find faster algorithms. In the simple cases of **Access** and **Swap**, the *optimal* algorithms for the considered bias are obtained. They are found by incorporating heuristics to the algorithm and storing constants in the initial values of the registers. The learned programs for these tasks are always in the first case of interpretability, this means that we can recover the most efficient algorithm from the learned weights.

Table 1: Average number of iterations required to solve instances of the problems for the original program, the best learned program and the ideal algorithm for the biased dataset. We also include the success rate of reaching a more efficient algorithm across multiple random restarts.

|  | Access | Increment | Swap | ListK | Addition | Sort |
|---|---|---|---|---|---|---|
| Generic | 6 | 40 | 10 | 18 | 20 | 38 |
| Learned | 4 | 16 | 6 | 11 | 9 | 18 |
| Ideal | 4 | 34 | 6 | 10 | 6 | 9.5 |
| Success Rate | 37 % | 84% | 27% | 19% | 12% | 74% |

While **ListK** and **Addition** have lower success rates, improvements between the original and learned algorithms are still significant. Both were initialised with iterative algorithms with $\mathcal{O}(n)$ complexities. They managed to find constant time $\mathcal{O}(1)$ algorithms to solve the given problems, making the runtime independent of the input. Achieving this means that the equivalence between the two approaches has been identified, similar to how optimising compilers operate. Moreover, on the **ListK** task, some learned programs corresponds to the second type of interpretability. Indeed these programs use soft jumps to condition the execution on the value of $k$. Even though these program would not generalise to other values of $k$, some learned programs for this task achieve a type one interpretability and a study of the learned algorithm reveal that they can generalise to *any* value of $k$.

Finally, the **Increment** task achieves an unexpected result. Indeed, it is able to outperform our best possible algorithm. By looking at the learned program, we can see that it is actually leveraging the possibility to perform soft writes over multiple elements of the memory at the same time to reduce its runtime. This is the only case where we see a learned program associated with the third type of interpretability. While our ideal algorithm would give a confidence of 1 on the output, this algorithm is unable to do so, but it has a high enough confidence of 0.9 to be considered a correct algorithm.

In practice, for all but the most simple tasks, we observe that further optimisation is possible, as some useless instructions remain present. Some transformations of the controller are indeed difficult to achieve through the local changes operated by the gradient descent algorithm. An analysis of these failure modes of our algorithm can be found in the supplementary materials. This motivates us to envision the use of approaches other than gradient descent to address these issues.

## 6  Discussion

The work presented here is a first step towards adaptive learning of programs. It opens up several interesting directions of future research. For exemple, the definition of efficiency that we considered in this paper is flexible. We chose to only look at the average number of operations executed to generate the output from the input. We leave the study of other potential measures such as Kolmogorov Complexity and `sloc`, to name a few, for future works.

As shown in the experiment section, our current method is very good at finding efficient solutions for simple programs. For more complex programs, only a solution close to the initialisation can be found. Even though training heuristics could help with the tasks considered here, they would likely not scale up to real applications. Indeed, the main problem we identified is that the gradient-descent based optimisation is unable to explore the space of programs effectively, by performing only local transformations. In future work, we want to explore different optimisation methods. One approach would be to mix global and local exploration to improve the quality of the solutions. A more ambitious plan would be to leverage the structure of the problem and use techniques from combinatorial optimisation to try and solve the original discrete problem.

## Acknowledgments

We would like to thank Siddharth Narayanaswamy and Diane Bouchacourt for helpful discussions and proof-reading the paper. This work was supported by the EPSRC, Leverhulme Trust, Clarendon Fund and the ERC grant ERC-2012-AdG 321162-HELIOS, EPSRC/MURI grant ref EP/N019474/1, EPSRC grant EP/M013774/1, EPSRC Programme Grant Seebibyte EP/M013774/1 and Microsoft Research PhD Scolarship Program.

## Footnotes

\*The first two authors contributed equally.

[1] https://github.com/albanD/adaptive-neural-compilation

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
