[Supplementary Material · supplementary_818.pdf]

# Adaptive Neural Compilation

## Supplementary Materials

## 1 Detailed Model Description

In this section, we are going to precisely define a non differentiable version model that we use in the main paper. This model can be seen as a recurrent network. Indeed, it takes as input an initial memory tape, performs a certain number of iterations and outputs a final memory tape. The memory tape is an array of $M$ cells, where a cell is an element holding a single integer value. The internal state of this recurrent model are the memory, the registers and the instruction register. The registers are another set of $R$ cells that are internal to the model. The instruction register is a single cell used in a specific way described later. These internal states are noted $\mathcal{M}^t = \{m_1^t, m_2^t, \ldots, m_M^t\}$, $\mathcal{R}^t = \{r_1^t, r_2^t, \ldots, r_R^t\}$ and $\mathcal{IR}^t$ for the memory, the registers and the instruction register respectively.

Figure 1 of the main paper describes in more detail how the different elements interact with each other. At each iteration, the Controller takes as input the value of the instruction register $\mathcal{IR}^t$ and outputs four values:

$$\mathrm{e}^t, \mathrm{a}^t, \mathrm{b}^t, \mathrm{o}^t = \mathrm{Controller}(\mathcal{IR}^t). \tag{1}$$

The first value $\mathrm{e}^t$ is used to select one of the instruction of the Machine to execute at this iteration. The second and third values $\mathrm{a}^t$ and $\mathrm{b}^t$ will identify which registers to use as the first and second argument for the selected instruction. The fourth value $\mathrm{o}^t$ identity the output register where to write the result of the executed instruction. The Machine then takes as input these four values and the internal state and computes the updated value of the internal state and a *stop* flag:

$$\mathcal{M}^{t+1}, \mathcal{R}^{t+1}, \mathcal{IR}^{t+1}, stop = \mathrm{Machine}(\mathcal{M}^t, \mathcal{R}^t, \mathcal{IR}^t, \mathrm{e}^t, \mathrm{a}^t, \mathrm{b}^t, \mathrm{o}^t). \tag{2}$$

The *stop* flag is a binary flag. When its value is 1, it means that the model will stop the execution and the current memory state will be returned.

**The Machine** The machine is a deterministic function that increments the instruction register and executes the command given by the Controller to update the current internal state. The set of instructions that can be executed by the Machine can be found in Figure 1b. Each instruction takes two values as arguments and returns a value. Additionally, some of these instructions have side effects. This mean that they do not just output a value, they perform another task. This other task can be for example to modify the content of the memory. All the considered side effects can be found in Figure 1b. By convention, instructions that don't have a value to return and that are used only for their side-effect will return a value of 0.

**The Controller** The Controller is a function that takes as input a single value and outputs four different values. The Controller's internal parameters, the initial values for the registers and the initial value of the instruction register define uniquely a given Controller.

The usual choice in the literature is to use an LSTM network[1, 2, 3] as controller. Our choice was to instead use a simpler model. Indeed, our Controller associates a command

to each possible value of the instruction register. Since the instruction register's value will increase by one at each iteration, this will enforce the Controller to encode in its weights what to do at each iteration. If we were using a recurrent controller the same instruction register could potentially be associated to different sets of outputs and we would lose this one to one mapping.

To make this clearer, we first rewrite the instruction register as an indicator vector with a 1 at the position of its value:

$$I_i = \begin{cases} 1 & \text{if } i = IR^t \\ 0 & \text{otherwise} \end{cases}. \tag{3}$$

In this case, we can write a single output $a^t$ of the Controller as the result of a linear function of $I$:

$$\mathtt{a}^{\mathtt{t}} = \mathtt{W_a} * I, \tag{4}$$

where $W_a$ is the 1xM matrix containing the value that need to be chosen as first arguments for each possible value of the instruction register and $*$ represent a matrix vector multiplication.

## 1.1 Mathematical details of the differentiable model

In order to make the model differentiable, every value and every choice are replaced by probability distributions over the possible choices. The main paper introduces how, using convex combinations of probability, the execution of the Machine is made differentiable. We present here the mathematical formulation of this procedure for the case of the side-effects, which was eluded in the paper for space reasons.

**STOP** In the discrete model, the execution is halted when the STOP instruction is executed. However, in the differentiable model, the STOP instruction may be executed with a probability smaller than 1. To take this into account, when executing the model, we keep track of the probability that the program should have terminated before this iteration based on the probability associated to the STOP instruction at each iteration. Once this probability goes over a threshold $\eta_{\text{stop}} \in ]0, 1]$, the execution is halted.

**READ** The mechanism is entirely the same as the one used to compute the arguments based on the registers and a probability distribution over the registers.

**JEZ** We note $\boldsymbol{\mathcal{IR}}_{jez}^{t+1}$ and $\boldsymbol{\mathcal{IR}}_{njez}^{t+1}$ the new value of $\boldsymbol{\mathcal{IR}}^t$ if we had respectively executed or not the JEZ instruction. We also have $e_{jez}^t$ the probability of executing this instruction at iteration $t$. The new value of the instruction register is:

$$\boldsymbol{\mathcal{IR}}^{t+1} = \boldsymbol{\mathcal{IR}}_{njez}^{t+1} \cdot (1 - e_{jez}^t) + \boldsymbol{\mathcal{IR}}_{jez}^{t+1} \cdot e_{jez}^t \tag{5}$$

$\boldsymbol{\mathcal{IR}}_{jez}^{t+1}$ is himself computed based on several probability distribution. If we consider that the instruction JEZ is executed with probabilistic arguments **cond** and **label**, its value is given by

$$\boldsymbol{\mathcal{IR}}_{jez}^{t+1} = \mathbf{label} \cdot \text{cond}_0 + \text{INC}(\boldsymbol{\mathcal{IR}}^t) \cdot (1 - \text{cond}_0) \tag{6}$$

With a probability equals to the one that the first argument is equal to zero, the new value of $\boldsymbol{\mathcal{IR}}^t$ is **label**. With the complement, it is equal to the incremented version of its current value, as the machine automatically increments the instruction register.

**WRITE** The mechanism is fairly similar to the one of the JEZ instruction.

We note $\mathbf{M}_{WRITE}^{t+1}$ and $\mathbf{M}_{nWRITE}^{t+1}$ the new value of $\mathbf{M}^t$ if we had respectively executed or not the WRITE instruction. We also have $e_{write}^t$ the probability of executing this instruction at iteration $t$. The new value of the memory matrix register is:

$$\mathbf{M}^{t+1} = \mathbf{M}_{nWRITE}^{t+1} \cdot (1 - e_{write}^t) + \mathbf{M}_{WRITE}^{t+1} \cdot e_{WRITE}^t \tag{7}$$

As with the JEZ instruction, the value of $\mathbf{M}_{WRITE}^{t+1}$ is dependent on the two probability distribution given as input: **addr** and **val**. The probability that the $i$-th cell of the memory

tape contains the value $j$ after the update is:

$$M_{i,j}^{t+1} = \text{addr}_i \cdot \text{val}_j + (1 - \text{addr}_i) \cdot M_{i,j}^t \tag{8}$$

Note that this can done using linear algebra operations so as to update everything in one global operation.

$$\mathbf{M}^{t+1} = \left( ((\mathbf{1} - \mathbf{addr})\mathbf{1}^T) \otimes \mathbf{M}^t \right) + (\mathbf{addr\ val}^T) \tag{9}$$

## 2    Specification of the loss

This loss contains four terms that will balance the correctness of the learnt algorithm, proper usage of the stop signal and speed of the algorithms. The parameters defining the models are the weight of the Controller's function and the initial value of the registers. When running the model with the parameters $\theta$, we consider that the execution ran for $T$ time steps. We consider the memory to have a size $M$ and that each number can be an integer between 0 and $M - 1$. $\mathbf{M}^t$ was the state of the memory at the $t$-th step. $\mathbf{T}$ and $\mathbf{C}$ are the target memory and the 0-1 mask of the elements we want to consider. All these elements are matrices where for example $\mathbf{M}_{i,j}^t$ is the probability of the $i$-th entry of the memory to take the value $j$ at the step $t$. We also note $p_{\text{stop},t}$ the probability outputted by the Machine that it should have stopped before iteration $t$.

**Correctness**    The first term corresponds to the correctness of the given algorithm. For a given input, we have the expected output and a mask. The mask allows us to know which elements in the memory we should consider when comparing the solutions. For the given input, we will compare the values specified by the mask of the expected output with the final memory tape provided by the execution. We compare them with the $\mathcal{L}_2$ distance in the probability space. Using the notations from above, we can write this term as:

$$L_c(\theta) = \sum_{i,j} \mathbf{C}_{i,j} (\mathbf{M}_{i,j}^T(\theta) - \mathbf{T}_{i,j})^2. \tag{10}$$

If we optimised only this first term, nothing would encourage the learnt algorithm to use the STOP instruction and halt as soon as it finished.

**Halting**    To prevent programs to take an infinite amount of time without stopping, we defined a maximum number of iterations $T_{max}$ after which the execution is halted. During training, we also add a penalty if the Controller didn't halt before this limit:

$$L_{sT_{max}}(\theta) = (1 - p_{\text{stop}-T}(\theta)) \cdot [T == T_{max}] \tag{11}$$

**Efficiency**    If we consider only the above mentioned losses, the program will make sure to halt by itself but won't do it as early as possible. We incentivise this behaviour by penalising each iteration taken by the program where it does not stop:

$$L_t(\theta) = \sum_{t \in [1, T-1]} (1 - p_{\text{stop},t}(\theta)). \tag{12}$$

**Confidence**    Moreover, we want the algorithm to have a good confidence to stop when it has found the correct output. To do so, we add the following term which will penalise probability of stopping if the current state of the memory is not the expected one:

$$L_s t(\theta) = \sum_{t \in [2, T]} \sum_{i,j} (p_{\text{stop},t}(\theta) - p_{\text{stop},t-1}(\theta)) \mathbf{C}_{i,j} (\mathbf{M}_{i,j}^t(\theta) - \mathbf{T}_{i,j})^2. \tag{13}$$

The increase in probability $(p_{\text{stop},t} - p_{\text{stop},t-1})$ corresponds to the probability of stopping exactly at iteration $t$. So, this is equivalent to the expected error made.

**Total loss**    The complete loss that we use is then the following:

$$L(\theta) = \alpha L_c(\theta) + \beta L_{sT_{max}}(\theta) + \gamma L_s t(\theta) + \delta L_t(\theta). \tag{14}$$

# 3   Distributed representation of the program

For the most of out experiments, the learned weights are fully interpretable as they first in the first type of interpretability. However, in some specific cases, under the pressure of our loss encouraging a smaller number of iterations, an interesting behavior emerges.

**Remarks**   It is interesting to note that the decompiled version is not straightforward to interpret. Indeed when we reach a program that has non Dirac-delta distributions in its weights, we cannot perform the inverse of the one-to-one mapping performed by the compiler. In fact, it relies on this blurriness to be able to execute the program with a smaller number of instruction. Notably, by having some blurriness on the JEZ instruction, the program can hide additional instructions, by creating a distributed state. We now explain the mechanism used to achieve this.

**Creating a distributed state**   Consider the following program and assume that the initial value of $\mathcal{IR}$ is 0:

Initial Registers:
$R_1 = 0; R_2 = 1; R_3 = 4, R_4 = 0$

Program:
$0 : R_1 = \textbf{READ} \ (R_1, R_4)$
$1 : R_4 = \textbf{JEZ} \ (R_1, R_3)$
$2 : R_4 = \textbf{WRITE}(R_1, R_1)$
$3 : R_4 = \textbf{WRITE}(R_1, R_3)$

If you take this program and execute it for three iterations, it will: read the first value of the tape into $R_1$. Then, if this value is zero, it will jump to State 4, otherwise it will just increment $\mathcal{IR}$. This means that depending on the value that was in $R_1$, the next instruction that will be executed will be different (in this case, the difference between State 3 and State 4 is which registers they will be writing from). This is our standard way of implementing conditionals.

Imagine that, after learning, the second instruction in our example program has 0.5 probability of being a JEZ and 0.5 probability of being a ZERO. If the content of $R_1$ is a zero, according to the JEZ, we should jump to State 4, but this instruction is executed with a probability of 0.5. We also have 0.5 probability of executing the ZERO instruction, which would lead to State 3.

Therefore, $\mathcal{IR}$ is not a Dirac-delta distribution anymore but points to State 3 with probability 0.5 and State 4 with probability 0.5.

**Exploiting a distributed state**   To illustrate, we will discuss how the Controller computes **a** for a model with 3 registers. The Table 1 show an example of some weights for such a controller.

|          | $R_1$ | $R_2$ | $R_3$ |
|----------|-------|-------|-------|
| State 1  | 20    | 5     | -20   |
| State 2  | -20   | 5     | 20    |

Table 1:   Controller Weights

If we are in State 1, the output of the controller is going to be
$$out = \text{softmax}([20, 5, -20]) = [0.9999..., 3e^{-7}, 4e^{-18}] \tag{15}$$
If we are in State 2, the output of the controller is going to be
$$out = \text{softmax}([-20, 5, 20]) = [4e^{-18}, 3e^{-7}, 0.9999...] \tag{16}$$

In both cases, the output of the controller is therefore going to be almost discrete. In State 1, $R_1$ would be chosen and in State 2, $R_3$ would be chosen.

However, in the case where we have a distributed state with probability 0.5 over State 1 and 0.5 over State 2, the output would be:

$$\begin{aligned} out &= \text{softmax}(0.5 * [-20, 5, 20] + 0.5[20, 5, -20]) \\ &= \text{softmax}([0, 10, 0]) \\ &= [4e^{-5}, 0.999, 4e^{-5}]. \end{aligned} \quad (17)$$

Note that the result of the distributed state is actually different from the result of the discrete states. Moreover it is still a discrete choice of the second register.

Because this program contains distributed elements, it is not possible to perform the one-to-one mapping between the weights and the lines of code. Though every instruction executed by the program, except for the JEZ, are binary. This means that this model can be translated to a regular program that will take exactly the same runtime, but will require more lines of codes than the number of lines in the matrix.

## 4   Alternative Learning Strategies

A critique that can be made to this method is that we will still initialise close to a local minimum. Another approach might be to start from a random initialisation but adding a penalty on the value of the weights such that they are encourage to be close to the generic algorithm. This can be seen as $\mathcal{L}_2$ regularisation but instead of pushing the weights to 0, we push then with the value corresponding to the generic algorithm. If we start with a very high value of this penalty but use an annealing schedule where its importance is very quickly reduced, this is going to be equivalent to the previous method.

## 5   Possible Extension

### 5.1   Making objective function differentiable

These experiments showed that we can transform any program that perform a mapping between an input memory tape to an output memory tape to a set of parameters and execute it using our model. The first point we want to make here is that this means that we take any program and transform it into a differentiable function easily. For example, if we want to learn a model that given a graph and two nodes a and b, will output the list of nodes to go through to go from a to b in the shortest amount of time. We can easily define the loss of the length of the path outputted by the model. Unfortunately, the function that computes this length from the set of nodes is not differentiable. Here we could implement this function in our model and use it between the prediction of the model and the loss function to get an end to end trainable system.

### 5.2   Beyond mimicking and towards open problems

It would even be possible to generalise our learning procedure to more complex problems for which we don't have a ground truth output. For example, we could consider problems where the exact answer for a given input is not computable or not unique. If the goodness of a solution can be computed easily, this value could be used as training objective. Any program giving a solution could be used as initialisation and our framework would improve it, making it generate better solutions.

## 6   Example tasks

This section will present the programs that we use as initialisation for the experiment section.

### 6.1   Access

In this task, the first element in the memory is a value $k$. Starting from the second element, the memory contains a zero-terminated list. The goal is to access the $k$-th element in the list that is zero-indexed. The program associated with this task can be found in Listing 1.

```
1  var k = 0
2  k = READ(0)
3  k = INC(k)
4  k = READ(k)
5  WRITE(0, k)
6  STOP()
```

Listing 1: Access Task

| Example input: | 6 | 9 | 1 | 2 | 7 | 9 | 8 | 1 | 3 | 5 |
|---|---|---|---|---|---|---|---|---|---|---|
| Output: | 1 | 9 | 1 | 2 | 7 | 9 | 8 | 1 | 3 | 5 |

## 6.2 Copy

In this task, the first element in the memory is a pointer $p$. Starting from the second element, the memory contains a zero-terminated list. The goal is to copy this list at the given pointer. The program associated with this task can be found in Listing 2.

```
1   var read_addr = 0
2   var read_value = 0
3   var write_addr = 0
4
5   write_addr = READ(0)
6   l_loop: read_value = READ(read_addr)
7   JEZ(read_value, l_stop)
8   WRITE(write_addr, read_value)
9   read_addr = INC(read_addr)
10  write_addr = INC(write_addr)
11  JEZ(0, l_loop)
12
13  l_stop: STOP()
```

Listing 2: Copy Task

| Example input: | 9 | 11 | 3 | 1 | 5 | 14 | 0 | 0 | 0 | 0 | 0 | 0 | 0 | 0 | 0 |
|---|---|---|---|---|---|---|---|---|---|---|---|---|---|---|---|
| Output: | 9 | 11 | 3 | 1 | 5 | 14 | 0 | 0 | 0 | 11 | 3 | 1 | 5 | 14 | 0 |

## 6.3 Increment

In this task, the memory contains a zero-terminated list. The goal is to increment each value in the list by 1. The program associated with this task can be found in Listing 3.

```
1   var read_addr = 0
2   var read_value = 0
3
4   l_loop: read_value = READ(read_addr)
5   JEZ(read_value, l_stop)
6   read_value = INC(read_value)
7   WRITE(read_addr, read_value)
8   read_addr = INC(read_addr)
9   JEZ(0, l_loop)
10
11  l_stop: STOP()
```

Listing 3: Increment Task

| Example input: | 1 | 2 | 2 | 3 | 0 | 0 | 0 |
|---|---|---|---|---|---|---|---|
| Output: | 2 | 3 | 3 | 4 | 0 | 0 | 0 |

## 6.4 Reverse

In this task, the first element in the memory is a pointer $p$. Starting from the second element, the memory contains a zero-terminated list. The goal is to copy this list at the given pointer in the reverse order. The program associated with this task can be found in Listing 4.

```
1  var read_addr = 0
2  var read_value = 0
3  var write_addr = 0
4
5  write_addr = READ(write_addr)
6  l_count_phase: read_value = READ(read_addr)
7  JEZ(read_value, l_copy_phase)
8  read_addr = INC(read_addr)
9  JEZ(0, l_count_phase)
10
11 l_copy_phase: read_addr = DEC(read_addr)
12 JEZ(read_addr, l_stop)
13 read_value = READ(read_addr)
14 WRITE(write_addr, read_value)
15 write_addr = INC(write_addr)
16 JEZ(0, l_copy_phase)
17
18 l_stop: STOP()
```

Listing 4: Reverse Task

| Example input: | 5 | 7 | 2 | 13 | 14 | 0 | 0 | 0 | 0 | 0 | 0 | 0 | 0 | 0 | 0 |
|---|---|---|---|---|---|---|---|---|---|---|---|---|---|---|---|
| Output: | 5 | 7 | 2 | 13 | 14 | 14 | 13 | 2 | 7 | 0 | 0 | 0 | 0 | 0 | 0 |

## 6.5 Permutation

In this task, the memory contains two zero-terminated list one after the other. The first contains a set of indices. the second contains a set of values. The goal is to fill the first list with the values in the second list at the given index. The program associated with this task can be found in Listing 5.

```
1  var read_addr = 0
2  var read_value = 0
3  var write_offset = 0
4
5  l_count_phase: read_value = READ(write_offset)
6  write_offset = INC(write_offset)
7  JEZ(read_value, l_copy_phase)
8  JEZ(0, l_count_phase)
9
10 l_copy_phase: read_value = DEC(read_addr)
11 JEZ(read_value, l_stop)
12 read_value = ADD(write_offset, read_value)
13 read_value = READ(read_value)
14 WRITE(read_addr, read_value)
15 read_addr = INC(read_addr)
16 JEZ(0, l_copy_phase)
17 l_stop: STOP()
```

Listing 5: Permutation Task

| Example input: | 2 | 1 | 3 | 0 | 13 | 4 | 6 | 0 | 0 | 0 | 0 | 0 | 0 | 0 | 0 |
|---|---|---|---|---|---|---|---|---|---|---|---|---|---|---|---|
| Output: | 4 | 13 | 6 | 0 | 13 | 4 | 6 | 0 | 0 | 0 | 0 | 0 | 0 | 0 | 0 |

## 6.6 Swap

In this task, the first two elements in the memory are pointers $p$ and $q$. Starting from the third element, the memory contains a zero-terminated list. The goal is to swap the elements pointed by $p$ and $q$ in the list that is zero-indexed. The program associated with this task can be found in Listing 6.

```
1  var p = 0
2  var p_val = 0
3  var q = 0
4  var q_val = 0
5
6  p = READ(0)
7  q = READ(1)
8  p_val = READ(p)
9  q_val = READ(q)
10 WRITE(q, p_val)
11 WRITE(p, q_val)
12 STOP()
```

Listing 6: Swap Task

| Example input: | 1 | 3 | 7 | 6 | 7 | 5 | 2 | 0 | 0 | 0 |
|---|---|---|---|---|---|---|---|---|---|---|
| Output: | 1 | 3 | 7 | 5 | 7 | 6 | 2 | 0 | 0 | 0 |

## 6.7 ListSearch

In this task, the first three elements in the memory are a pointer to the head of the linked list, the value we are looking for $v$ and a pointer to a place in memory where to store the result. The rest of the memory contains the linked list. Each element in the linked list is two values, the first one is the pointer to the next element, the second is the value contained in this element. By convention, the last element in the list points to the address 0. The goal is to return the pointer to the first element whose value is equal to $v$. The program associated with this task can be found in Listing 7.

```
1  var p_out = 0
2  var p_current = 0
3  var val_current = 0
4  var val_searched = 0
5
6  val_searched = READ(1)
7  p_out = READ(2)
8  l_loop: p_current = READ(p_current)
9  val_current = INC(p_current)
10 val_current = READ(val_current)
11 val_current = SUB(val_current, val_searched)
12 JEZ(val_current, l_stop)
13 JEZ(0, l_loop)
14 l_stop: WRITE(p_out, p_current)
15 STOP()
```

Listing 7: ListSearch Task

| Example input: | 11 | 10 | 2 | 9 | 4 | 3 | 10 | 0 | 6 | 7 | 13 | 5 | 12 | 0 | 0 |
|---|---|---|---|---|---|---|---|---|---|---|---|---|---|---|---|
| Output: | 11 | 10 | 5 | 9 | 4 | 3 | 10 | 0 | 6 | 7 | 13 | 5 | 12 | 0 | 0 |

## 6.8 ListK

In this task, the first three elements in the memory are a pointer to the head of the linked list, the number of hops we want to do $k$ in the list and a pointer to a place in memory where to store the result. The rest of the memory contains the linked list. Each element in the linked list is two values, the first one is the pointer to the next element, the second is the value contained in this element. By convention, the last element in the list points to the address 0. The goal is to return the value of the $k$-th element of the linked list. The program associated with this task can be found in Listing 8.

```
1   var p_out = 0
2   var p_current = 0
3   var val_current = 0
4   var k = 0
5
6   k = READ(1)
7   p_out = READ(2)
8   l_loop: p_current = READ(p_current)
9   k = DEC(k)
10  JEZ(k, l_stop)
11  JEZ(0, l_loop)
12  l_stop: p_current = INC(p_current)
13  p_current = READ(p_current)
14  WRITE(p_out, p_current)
15  STOP()
```

Listing 8: ListK Task

| Example input: | 3 | 2 | 2  | 9 | 15 | 0 | 0 | 0 | 1 | 15 | 17 | 7 | 13 | 0 | 0 | 11 |
|----------------|---|---|----|---|----|---|---|---|---|----|----|---|----|---|---|----|
| Output:        | 3 | 2 | 17 | 9 | 15 | 0 | 0 | 0 | 1 | 15 | 17 | 7 | 13 | 0 | 0 | 11 |

| 10 | 0 | 0 | 0 |
|----|---|---|---|
| 10 | 0 | 0 | 0 |

## 6.9 Walk BST

In this task, the first two elements in the memory are a pointer to the head of the BST and a pointer to a place in memory where to store the result. Starting at the third element, there is a zero-terminated list containing the instructions on how to traverse in the BST. The rest of the memory contains the BST. Each element in the BST has three values, the first one is the value of this node, the second is the pointer to the left node and the third is the pointer to the right element. By convention, the leafs points to the address 0. The goal is to return the value of the node we get at after following the instructions. The instructions are 1 or 2 to go respectively to the left or the right. The program associated with this task can be found in Listing 9.

```
1   var p_out = 0
2   var p_current = 0
3   var p_instr = 0
4   var instr = 0
5
6   p_current = READ(0)
7   p_out = READ(1)
8   instr = READ(2)
9
10  l_loop: JEZ(instr, l_stop)
11  p_current = ADD(p_current, instr)
12  p_current = READ(p_current)
13  p_instr = INC(p_instr)
14  JEZ(0, l_loop)
15
16  l_stop: p_current = READ(p_current)
17  WRITE(p_out, p_current)
18  STOP()
```

Listing 9: WalkBST Task

| Example input: | 12 | 1 | 1 | 2 | 0 | 0 | 15 | 0 | 9 | 23 | 0 | 0 | 11 | 15 | 6 |
|---|---|---|---|---|---|---|---|---|---|---|---|---|---|---|---|
| Output: | 12 | 10 | 1 | 2 | 0 | 0 | 15 | 0 | 9 | 23 | 0 | 0 | 11 | 15 | 6 |

| 8 | 0 | 24 | 0 | 0 | 0 | 0 | 0 | 0 | 10 | 0 | 0 | 0 | 0 | 0 |
|---|---|---|---|---|---|---|---|---|---|---|---|---|---|---|
| 8 | 0 | 24 | 0 | 0 | 0 | 0 | 0 | 0 | 10 | 0 | 0 | 0 | 0 | 0 |

## 6.10   Merge

In this task, the first three elements in the memory are pointers to respectively, the first list, the second list and the output. The two lists are zero-terminated sorted lists. The goal is to merge the two lists into a single sorted zero-terminated list that starts at the output pointer. The program associated with this task can be found in Listing 10.

```
1   var p_first_list = 0
2   var val_first_list = 0
3   var p_second_list = 0
4   var val_second_list = 0
5   var p_output_list = 0
6   var min = 0
7
8   p_first_list = READ(0)
9   p_second_list = READ(1)
10  p_output_list = READ(2)
11
12  l_loop: val_first_list = READ(p_first_list)
13  val_second_list = READ(p_second_list)
14  JEZ(val_first_list, l_first_finished)
15  JEZ(val_second_list, l_second_finished)
16  min = MIN(val_first_list, val_second_list)
17  min = SUB(val_first_list, min)
18  JEZ(min, l_first_smaller)
19
20  WRITE(p_output_list, val_first_list)
21  p_output_list = INC(p_output_list)
22  p_first_list = INC(p_first_list)
23  JEZ(0, l_loop)
24
25  l_first_smaller: WRITE(p_output_list, val_second_list)
26  p_output_list = INC(p_output_list)
27  p_second_list = INC(p_second_list)
28  JEZ(0, l_loop)
29
30  l_first_finished: p_first_list = ADD(p_second_list, 0)
31  val_first_list = ADD(val_second_list, 0)
32
33  l_second_finished: WRITE(p_output_list, val_first_list)
34  p_first_list = INC(p_first_list)
35  p_output_list = INC(p_output_list)
36  val_first_list = READ(p_first_list)
37  JEZ(val_first_list, l_stop)
38  JEZ(0, l_second_finished)
39
40  l_stop: STOP()
```

Listing 10: Merge Task

| Example input: | 3 | 8 | 11 | 27 | 17 | 16 | 1 | 0 | 29 | 26 | 0 | 0 | 0 | 0 | 0 |
|---|---|---|---|---|---|---|---|---|---|---|---|---|---|---|---|
| Output: | 3 | 8 | 11 | 27 | 17 | 16 | 1 | 0 | 29 | 26 | 0 | 29 | 27 | 26 | 17 |

| 0 | 0 | 0 | 0 | 0 | 0 | 0 | 0 | 0 | 0 | 0 | 0 | 0 | 0 | 0 |
|---|---|---|---|---|---|---|---|---|---|---|---|---|---|---|
| 16 | 1 | 0 | 0 | 0 | 0 | 0 | 0 | 0 | 0 | 0 | 0 | 0 | 0 | 0 |

## 6.11   Dijkstra

In this task, we are provided with a graph represented in the input memory as follow. The first element is a pointer $p_out$ indicating where to write the results. The following elements contain a zero-terminated array with one entry for each vertex in the graph. Each entry is a pointer to a zero-terminated list that contains a pair of values for each outgoing edge of the considered node. Each pair of value contains first the index in the first array of the child node and the second value contains the cost of this edge. The goal is to write a zero-terminated list at the address provided by $p_out$ that will contain the value of the shortest path from the

first node in the list to this node. The program associated with this task can be found in Listings 11 and 12.

```
1   var min = 0
2   var argmin = 0
3
4   var p_out = 0
5   var p_out_temp = 0
6   var p_in = 1
7   var p_in_temp = 1
8
9   var nnodes = 0
10
11  var zero = 0
12  var big = 99
13
14  var tmp_node = 0
15  var tmp_weight = 0
16  var tmp_current = 0
17  var tmp = 0
18
19  var didsmth = 0
20
21  p_out = READ(p_out)
22  p_out_temp = ADD(p_out, zero)
23
24  tmp_current = INC(zero)
25  l_loop_nnodes:tmp = READ(p_in_temp)
26  JEZ(tmp, l_found_nnodes)
27  WRITE(p_out_temp, big)
28  p_out_temp = INC(p_out_temp)
29  WRITE(p_out_temp, tmp_current)
30  p_out_temp = INC(p_out_temp)
31  p_in_temp = INC(p_in_temp)
32  nnodes = INC(nnodes)
33  JEZ(zero, l_loop_nnodes)
34
35  l_found_nnodes:WRITE(p_out, zero)
36  JEZ(zero, l_find_min)
37  l_min_return:p_in_temp = ADD(p_in, argmin)
38  p_in_temp = READ(p_in_temp)
39
40  l_loop_sons:tmp_node = READ(p_in_temp)
41  JEZ(tmp_node, l_find_min)
42  tmp_node = DEC(tmp_node)
43  p_in_temp = INC(p_in_temp)
44  tmp_weight = READ(p_in_temp)
45  p_in_temp = INC(p_in_temp)
46
47  p_out_temp = ADD(p_out, tmp_node)
48  p_out_temp = ADD(p_out_temp, tmp_node)
49  tmp_current = READ(p_out_temp)
50  tmp_weight = ADD(min, tmp_weight)
51
52  tmp = MIN(tmp_current, tmp_weight)
53  tmp = SUB(tmp_current, tmp)
54  JEZ(tmp, l_loop_sons)
55  WRITE(p_out_temp, tmp_weight)
56  JEZ(zero, l_loop_sons)
```

Listing 11: Dijkstra Algorithm (Part 1)

```
57  l_find_min:p_out_temp = DEC(p_out)
58  tmp_node = DEC(zero)
59  min = ADD(big, zero)
60  argmin = DEC(zero)
61
62  l_loop_min:p_out_temp = INC(p_out_temp)
63  tmp_node = INC(tmp_node)
64  tmp = SUB(tmp_node, nnodes)
65  JEZ(tmp, l_min_found)
66
67  tmp_weight = READ(p_out_temp)
68
69  p_out_temp = INC(p_out_temp)
70  tmp = READ(p_out_temp)
71  JEZ(tmp, l_loop_min)
72
73  tmp = MAX(min, tmp_weight)
74  tmp = SUB(tmp, tmp_weight)
75  JEZ(tmp, l_loop_min)
76  min = ADD(tmp_weight, zero)
77  argmin = ADD(tmp_node, zero)
78  JEZ(zero, l_loop_min)
79
80  l_min_found:tmp = SUB(min, big)
81  JEZ(tmp, l_stop)
82  p_out_temp = ADD(p_out, argmin)
83  p_out_temp = ADD(p_out_temp, argmin)
84  p_out_temp = INC(p_out_temp)
85  WRITE(p_out_temp, zero)
86  JEZ(zero, l_min_return)
87
88  l_stop:STOP()
```
Listing 12: Dijkstra Algorithm (Part 2)

*Example omitted for space reasons*

# 7 Learned optimisation: Case study

Here we present an analysis of the optimisation achieved by the ANC. We take the example of the **ListK** task and study the difference between the learned program and the initialisation used.

## 7.1 Representation

The representation chosen is under the form of the intermediary representation described in Figure (2b) of the main paper. Based on the parameters of the Controller, we can recover the approximate representation described in Figure (2b) of the main paper: 1For each possible "discrete state" of the instruction register, we can compute the commands outputted by the controller. We report the most probable value for each distribution, as well as the probability that the compiler would assign to this value. If no value has a probability higher than 0.5, we only report a neutral token (R-, -, NOP).

## 7.2 Biased ListK

Figure 1 represents the program that was used as initialisation to the optimisation problem. This is the direct result from the compilation performed by the Neural Compiler of the

program described in Listing 8. A version with a probability of 1 for all necessary instructions would have been easily obtained but not amenable to learning.

Figure 2 similarly describes the program that was obtained after learning.

As a remainder, the bias introduced in the ListK task is that the linked list is well organised in memory. In the general case, the element could be in any order. An input memory tape to the problem of asking for the third element in the linked list containing $\{4, 5, 6, 7\}$ would be:

| 9 | 3 | 2 | 0 | 0 | 11 | 5 | 0 | 7 | 5 | 4 | 7 | 6 | 0 | 0 | 0 | 0 | 0 | 0 | 0 |
|---|---|---|---|---|----|---|---|---|---|---|---|---|---|---|---|---|---|---|---|

or

| 5 | 3 | 2 | 0 | 0 | 7 | 4 | 15 | 5 | 0 | 7 | 0 | 0 | 0 | 0 | 9 | 6 | 0 | 0 | 0 |
|---|---|---|---|---|---|---|----|---|---|---|---|---|---|---|---|---|---|---|---|

In the biased version of the task, all the elements are arranged in order and contiguously positioned on the tape. The only valid representation of this problems is:

| 3 | 3 | 2 | 5 | 4 | 7 | 5 | 9 | 6 | 0 | 7 | 0 | 0 | 0 | 0 | 0 | 0 | 0 | 0 | 0 |
|---|---|---|---|---|---|---|---|---|---|---|---|---|---|---|---|---|---|---|---|

## 7.3 Solutions

Because of the additional structure of the problem, the bias in the data, a more efficient algorithm to find the solution exists. Let us dive into the comparison of the two different solutions.

Both use their first two states to read the parameters of the given instance of the task. Which element of the list should be returned is read at line (0:) and where to write the returned value is read at line (1:). Step (2:) to (6:) are dedicated to putting the address of the $k$-th value of the linked list into the registers R1. Step (7:) to (9:) perform the same task in both solution: reading the value at the address contained in R1, writing it at the desired position and stopping.

The difference between the two programs lies in how they put the address of the $k$-th value into R1.

**Generic** The initialisation program, used for initialisation, works in the most general case so needs to perform a loop where it put the address of the next element in the linked list in R1 (2:), decrement the number of jumps remaining to be made (3:), checking whether the wanted element has been reached (4:) and going back to the start of the loop if not (5:). Once the desired element is reached, R1 is incremented so as to point on the value of the linked list element.

**Specific** On the other hand, in the biased version of the problem, the position of the desired value can be analytically determined. The function parameters occupy the first three cells of the tape. After those, each element of the linked list will occupy two cells (one for the pointer to the next address and one for the value). Therefore, the address of the desired value is given by

$$\begin{aligned} \text{R1} &= 3 + (2*(k-1)+1) - 1 + 1 \\ &= 3 + 2*k - 1 \end{aligned} \tag{18}$$

(the -1 comes from the fact that the address are 0-indexed and the final +1 from the fact that we are interested in the position of the value and not of the pointer.)

The way this is computed is as follows:

- R1 $= 3 + k$ by adding the constant 3 to the registers R2 containing K. (2:)

- R2 $= k - 1$ (3:)

- R1 $= 3 + 2*k - 1$ by adding the now reduced value of R2. (6:)

The algorithm implemented by the learned version is therefore much more efficient for the biased dataset, due to its capability to ignore the loop.

### 7.4 Failure analysis

An observation that can be made is that in the learned version of the program, Step (4:) and (5:) are not contributing to the algorithms. They execute instructions that have no side effect and store the results into the registers R7 that is never used later in the execution.

The learned algorithm could easily be more efficient by not performing these two operations. However, such an optimisation, while perhaps trivial for a standard compiler, capable of detecting unused values, is fairly hard for our optimisers to discover. Because we are only doing gradient descent, the action of "moving some instructions earlier in the program" which would be needed here to make the useless instructions disappear, is fairly hard, as it involves modifying several rows of the program at once in a coherent manner.

```
 R1 = 0 (0.88)
 R2 = 1 (0.88)
 R3 = 2 (0.88)
 R4 = 6 (0.88)
 R5 = 0 (0.88)
 R6 = 2 (0.88)
 R7 = - (0.05)

 Initial State: 0 (0.88)

 0:  R2 (0.96)   = READ (0.93)   [ R2 (0.96)    , R- (0.14)    ]
 1:  R3 (0.96)   = READ (0.93)   [ R3 (0.96)    , R- (0.14)    ]
 2:  R1 (0.96)   = READ (0.93)   [ R1 (0.96)    , R- (0.14)    ]
 3:  R2 (0.96)   = DEC  (0.93)   [ R2 (0.96)    , R- (0.14)    ]
 4:  R7 (0.96)   = JEZ  (0.93)   [ R2 (0.96)    , R4 (0.96)    ]
 5:  R7 (0.96)   = JEZ  (0.93)   [ R5 (0.96)    , R6 (0.96)    ]
 6:  R1 (0.96)   = INC  (0.93)   [ R1 (0.96)    , R- (0.14)    ]
 7:  R1 (0.96)   = READ (0.93)   [ R1 (0.96)    , R- (0.14)    ]
 8:  R7 (0.96)   = WRIT (0.93)   [ R3 (0.96)    , R1 (0.96)    ]
 9:  R7 (0.96)   = STOP (0.93)   [ R- (0.14)    , R- (0.14)    ]
10:  R- (0.16)   = NOP  (0.11)   [ R- (0.16)    , R- (0.16)    ]
11:  R- (0.16)   = NOP  (0.11)   [ R- (0.18)    , R- (0.16)    ]
12:  R- (0.17)   = NOP  (0.1)    [ R- (0.17)    , R- (0.17)    ]
13:  R- (0.16)   = NOP  (0.11)   [ R- (0.17)    , R- (0.16)    ]
14:  R- (0.17)   = NOP  (0.11)   [ R- (0.16)    , R- (0.16)    ]
15:  R- (0.16)   = NOP  (0.1)    [ R- (0.16)    , R- (0.17)    ]
16:  R- (0.17)   = NOP  (0.11)   [ R- (0.16)    , R- (0.18)    ]
17:  R- (0.18)   = NOP  (0.1)    [ R- (0.18)    , R- (0.17)    ]
18:  R- (0.17)   = NOP  (0.1)    [ R- (0.16)    , R- (0.16)    ]
19:  R- (0.15)   = NOP  (0.11)   [ R- (0.16)    , R- (0.17)    ]
```

Figure 1: Initialisation used for the learning of the ListK task.

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

```
R1 = 3 (0.99)
R2 = 1 (0.99)
R3 = 2 (0.99)
R4 = 10 (0.53)
R5 = 0 (0.99)
R6 = 2 (0.99)
R7 = 7 (0.99)

Initial State: 0 (0.99)

0:   R2 (0.99)   = READ (0.99)   [ R2 (1)    , R1 (0.97)   ]
1:   R6 (0.99)   = READ (0.99)   [ R3 (0.99)  , R6 (0.5)   ]
2:   R1 (1)      = ADD  (0.99)   [ R1 (0.99)  , R2 (1)     ]
3:   R2 (1)      = DEC  (0.99)   [ R2 (1)     , R1 (0.99)  ]
4:   R7 (0.99)   = MAX  (0.99)   [ R2 (0.99)  , R1 (0.51)  ]
5:   R7 (0.99)   = INC  (0.99)   [ R6 (0.7)   , R1 (0.89)  ]
6:   R1 (1)      = ADD  (0.99)   [ R1 (0.99)  , R2 (1)     ]
7:   R1 (0.99)   = READ (0.99)   [ R1 (0.99)  , R1 (0.53)  ]
8:   R7 (0.99)   = WRIT (0.99)   [ R6 (1)     , R1 (0.99)  ]
9:   R7 (0.9)    = STOP (0.99)   [ R6 (0.98)  , R1 (0.99)  ]
10:  R2 (0.99)   = STOP (0.96)   [ R1 (0.52)  , R1 (0.99)  ]
11:  R1 (0.98)   = ADD  (0.73)   [ R4 (0.99)  , R2 (0.99)  ]
12:  R3 (0.98)   = ADD  (0.64)   [ R6 (0.99)  , R1 (0.99)  ]
13:  R3 (0.87)   = STOP (0.65)   [ R3 (0.52)  , R1 (0.99)  ]
14:  R3 (0.89)   = STOP (0.62)   [ R6 (0.99)  , R2 (0.62)  ]
15:  R3 (0.99)   = STOP (0.65)   [ R3 (0.99)  , R2 (0.71)  ]
16:  R3 (0.99)   = NOP  (0.45)   [ R6 (0.99)  , R1 (0.99)  ]
17:  R2 (0.99)   = INC  (0.56)   [ R6 (0.7)   , R1 (0.98)  ]
18:  R3 (0.99)   = STOP (0.65)   [ R3 (0.99)  , R1 (0.99)  ]
19:  R3 (0.98)   = STOP (0.98)   [ R2 (0.62)  , R1 (0.79)  ]
```

Figure 2: Learnt program for the listK task