[Reviews · NeurIPS 2016]

Reviewer 1

Summary

The authors of this paper present a compiler, that compiles from an assembly-like langauge to a Neural Turing Machine-like neural network. After compiling a program, one gets a set of parameters that -- when executed -- act exactly as the program. The claimed advantage of such compilation is that the architecture is now differentiable, so it can be trained. The authors present some results training simple programs, but (as admitted in the paper), they are not scalable.

Qualitative Assessment

This paper does 2 things: (1) presents a compiler from a simplified assembly langauge to a Neural Turing Machine-like architecture, and (2) documents an attempt to learn faster programs after (1) using the compiled weights. The problem is that (1) is a basic compilation technique, there is not much new in it. It would be interesting if (2) worked well, but that is not the case. The authors mention other approaches to learn algorithms, like the Neural Turing Machine or the Neural GPU, but then dismiss them saying they are "complex to train" (line 35). But then the authors consider tasks that have already been solved by these architectures, like copying sequences. In Table 1 (the only experimental results in the paper), the authors do not even compare to the existing baselines. They start from a pre-compiled network, but existing baselines can solve all the presented problems from random initialization -- why do we need the pre-compilation step? If it is to make programs faster, the paper lacks any results to show that. As for the presentation, it is acceptable, but lacks many details. For example, section 3.2 is quite long and describes a known simple technique, while section 4.1 lacks technical details in a place where the new model is introduced -- I suggest to change that. [Read the rebuttal, but the authors avoid comparing with existing architectures that can already learn *faster* algorithms for the presented tasks without any pre-compilation. This makes me think the presentation is misleading and made me lower my scores.]

Confidence in this Review

3-Expert (read the paper in detail, know the area, quite certain of my opinion)


Reviewer 2

Summary

This paper describes a "neural compiler" capable of (1) transforming assembly code into the weights of a recurrent controller network, working in tandem with a memory module and instruction registers to correctly run the program, and (2) optimize this network so as to run more efficiently on a particular distribution of program inputs. On six example algorithms, the proposed method is shown to be capable of inventing new algorithms that exploit bias in the input data, e.g. discovering that only the start of the arrays are unsorted, and ignoring the end portion. In some cases, the model is even shown to outperform "ideal" biased programs by performing soft writes over multiple memory elements.

Qualitative Assessment

I found this paper very enjoyable and thought-provoking. It is a strong addition to the growing body of work on neural program induction and optimization, and I would like to see it presented at NIPS. The proposed approach of *optimizing* a program with a neural network to make it run more efficiently (yet still correctly) on a distribution of inputs, rather than learning it from scratch from examples, appears to be quite effective. It is also reminiscent of the following work which I think should be cited: "Learning to Discover Efficient Mathematical Identities" Wojciech Zaremba, Karol Kurach, Rob Fergus Detailed comments / questions: - Equation 2 suggests that the output values are limited by the size of memory M. Does this limit the type of programs ANC can learn? - Are there any baselines in the non-neural program optimization literature that you could compare to on these tasks? - How might this approach be applied to visual programs, motor programs, or programs with perceptual inputs instead of symbols? This may be where neural methods can go where classical symbolic program optimization methods cannot. - When training on biased data, can the model learn algorithms that are *always correct* but faster than the initial algorithm in expectation? For example, could it learn to quickly check for exploitable bias, exploit it if possible, otherwise run the default?

Confidence in this Review

3-Expert (read the paper in detail, know the area, quite certain of my opinion)


Reviewer 3

Summary

This paper proposes an adaptive neural-compilation framework to address the problem of efficient program learning. The papers show that it is possible to compile programs written in a low-level language to a differentiable representation and can be optimised to make them efficient on a target distribution of inputs.

Qualitative Assessment

Typo: 233 that would not have be possible if ... About the success rate: From the results we can see that even with some very simple instances, the success rate is poor. It then concerns me that for such an application, the main purpose is to hopefully we can use it to optimize some complicated programs which has low efficiency (that's why we need the optimization), however, the proposed method will not be helping in this case, because it can only work for some very simple cases. Even though, the proposed method is very interesting and inspiring.

Confidence in this Review

2-Confident (read it all; understood it all reasonably well)


Reviewer 4

Summary

The paper explores a scheme where a low-level program is compiled into a differentiable representation, and the result is used as a starting point for a learning process optimizing a program for run-time efficiency on a selected distribution of inputs with the correctness constraints also applied only on the same distribution of inputs. In a number of cases a classical optimized program is recoverable from the differentiable representation found by the learning process, but not always. In some other cases the result of learning gives better performance than the optimal performance possible for a classical program (in those cases, obviously, a classical program is not recoverable from the continuous representation obtained).

Qualitative Assessment

Recently we have seen re-emergence of research into neural models of classical software. This line of research is now rapidly accelerating. For many of us, the single paper most associated with the starting point of this re-emergence was "Neural Turing Machines" by Graves et al., and the development was very rapid since then. It is important to explore various angles and viewpoints on this rapidly evolving research topic. The research presented in the paper under review is of interest on its own, but for many readers the key value will be provided by a new interesting angle of view this paper provides into differentiable models of classical software. This is where the most value from this paper for NIPS readership would probably come from. I liked the way the paper is written overall, and I benefited from reading it in its current form, but I found some aspects of it confusing and more difficult to the reader than they should be. Some of those confusing aspects are, unfortunately, close to the central part of technical presentation. I hope the authors will be able to improve the paper in those aspects, some of which I list below. It is not immediately obvious what the convex combination of registers is (formula 1, line 130). The natural interpretation is that it is a convex combination of delta-function probability distributions (distributions on the set of registers where a single register is taken with probability 1). But one also wonders whether it might be a convex combination of the values of registers (resulting in real values, rather than integers, and then using extension of operations listed in Figure 1b onto reals). In any case, if it is a convex combination of delta-function distributions, the paper should say so explicitly. If it is more subtle than that, the need for explanation is even higher. The formula 2, line 132 is just not interpretable for a reader, even after looking into the supplement. What those double indexes mean for "out" and for "arg" is just too difficult to guess correctly. A verbal explanation of what's going on in this formula is necessary. The role of soft-write and softmax in all this is not well explained (the reader can guess some of it from looking at the supplement). I think a much crisper explanation of those central aspects I've listed so far is necessary, preferably in the paper itself, rather than in the supplement. I am going to also list a few less significant aspects where improvement would be desirable. The term "instruction register (IR)" is somewhat confusing. First the reader might think, is this register going to actually store the machine command? Then the reader eventually realizes that this register is used simply as "program counter (PC)" (acceptable alternatives for this term seem to be "instruction pointer (IP)", "instruction address register (IAR)", or "instruction counter (IC)"). But the wording in line 104-105 does not say that the controller simply uses the "instruction register" to fetch the command from memory using the value of that counter and then parses the resulting command (which is what seems to actually be going on). Instead a wording is more general, and says that the controller computes the components of the command in some unspecified way. So the authors might have meant something else in mind in addition to the straightforward classical schema which they did end up implementing, and "instruction register" terminology might be hinting at those alternatives. In any case, it took some labor to understand that this was used as straightforward program counter, a pointer into the memory tape from which the command is fetched. In the differentiable version (Section 3.2) the authors seem to rely on the range of accessible interger values being the same as the size of the memory tape M. This range, which is the same as indices into M, should then be from 0 to M-1, line 97 nonwithstanding (because the paper relies on having 0, so indices for m's in line 97 should be adjusted for that). I understand that this is a convenient convention at the early stage of this research (one then can worry less about values of indices which don't point anywhere into the memory, etc), but is it really necessary? It does feel like a very artificial requirement. There are some minor problems with Figure 2c. The matrix (iv) is missing one row. In the matrix (i) "DEC" is not where it should be according to the order of Figure 1b. Finally the columns look OK on screen, but when I print, the first columns of matrices (1) and (iii) are only half-width, and the last columns of those matrices are one-and-a-half width. Reference [7] is missing the last name of the second author, Jimmy Ba. Instead, the first word of the paper title, "Adam", has migrated to the position of the second author's last name. Finally, "efficient program learning" on line 2 is quite ambiguous. The first interpretation which jumps into reader's mind is "efficient process of program learning". Only then one realizes that the authors has meant "learning the efficient programs" instead. I hope this would help to improve the text.

Confidence in this Review

3-Expert (read the paper in detail, know the area, quite certain of my opinion)


Reviewer 5

Summary

The paper presents a method of taking any program and rewriting it into a set of simpler instructions that have a differentiable counterpart. The paper shows that the resulting architecture can find very efficient solutions for some simple algorithmic tasks.

Qualitative Assessment

The ideas in the paper are simple and original. The main basic observation is that register machines can be made differentiable. Within the topic of program learning with neural networks, this observation is particularly nice. Also, in contrast with other program learning papers, the approach presented here does have a use case, which is the one of finding more efficient programs for certain problems given an empirical distribution on the inputs. All these ideas seem quite original and might inspire other avenues of research along the same path. From an experimental point of view, the results are somewhat limited to very simple tasks. As the authors note, it is not clear how these techniques could be extended to real world programs, partly due to the inherently local nature of parameter search via gradient descent. Nevertheless, the originality of the approach remains welcome.

Confidence in this Review

2-Confident (read it all; understood it all reasonably well)